# Six Species of *Phyllachora* with Three New Taxa on Grass from Sichuan Province, China

**DOI:** 10.3390/jof10080588

**Published:** 2024-08-19

**Authors:** Qi-Rong Sun, Xiu-Lan Xu, Zhen Zeng, Yu Deng, Feng Liu, Li-Ping Gao, Fei-Hu Wang, Ya-Qian Yan, Ying-Gao Liu, Chun-Lin Yang

**Affiliations:** 1College of Forestry, Sichuan Agricultural University, Chengdu 611130, China; 15911441516@163.com (Q.-R.S.); dy17711433642@163.com (Y.D.); liufeng981113@126.com (F.L.); scbzwfh@163.com (F.-H.W.); yyq18583272025@126.com (Y.-Q.Y.); 17612825070@163.com (Y.-G.L.); 2National Forestry and Grassland Administration Key Laboratory of Forest Resources Conservation and Ecological Safety on the Upper Reaches of the Yangtze River, College of Forestry, Sichuan Agricultural University, Chengdu 611130, China; 3Forestry Research Institute, Chengdu Academy of Agricultural and Forestry Sciences, Chengdu 611130, China; xuxiulanxxl@126.com (X.-L.X.); 15882215544@126.com (Z.Z.); 18116596692@139.com (L.-P.G.)

**Keywords:** three new taxa, *Poaceae*, *Phyllachora*, multigene phylogeny, taxonomy

## Abstract

*Phyllachora* (*Phyllachoraceae*, *Phyllachorales*) species are parasitic fungi with a wide global distribution, causing tar spots on plants. In this study, we describe three newly discovered species: *Phyllachora chongzhouensis*, *Phyllachora neidongensis*, and *Phyllachora huiliensis* from *Poaceae* in China. These species were characterized using morphological traits and multi-locus phylogeny based on the internal transcribed spacer region (ITS) with the intervening 5.8S rRNA gene, the large subunit of the rRNA gene (LSU), and the 18S ribosomal RNA gene (SSU). Three known species of *P. chloridis*, *P. graminis*, and *P. miscanthi* have also been redescribed, because, in reviewing the original references of *P. chloridis*, *P. graminis*, and *P. miscanthi*, these were found to be relatively old and in Chinese or abbreviated. In addition, the illustrations were simple. In molecular identification, the ITS sequence is short, while the ITS, LSU, and SSU are incomplete. Therefore, this study provides new important references for the redescription of three known species and provides further evidence for the identification of new taxa.

## 1. Introduction

The order *Phyllachorales* is a group of biotrophic, plant-parasitic fungi with high host specificity and a global distribution [1]. The species of *Phyllachorales* have a common name, ‘tar spot fungi’ [2], because they are usually leaf- or stem-inhabiting with shiny black stromata. Initially, the families *Phyllachoraceae* and *Phaeochoraceae* were classified within this order mostly based on their morphological characteristics and host preferences [3,4]. Then, a new family, *Telimenaceae*, typified with *Telimena erythrinae* Racib., was proposed to be separated from the family *Phyllachoraceae* with the aid of ancestral state reconstruction [5]. Subsequently, Guterres et al. [6] presented a monogeneric new family, *Neopolystigmataceae*, which appeared to be well supported within *Phyllachorales* in both maximum likelihood (ML) and Bayesian inference (BI) phylogenetic relationship analyses.

The family *Phyllachoraceae*, introduced by Theissen and Sydow [7], comprises approximately 54 genera [5]. *Phyllachora* is the largest genus; it houses 1084 species. (https://www.speciesfungorum.org/Names/Names.asp, accessed on 10 May 2024). The majority of these genera exhibit a pronounced affinity for plants, manifesting from sub-epidermal to extensive intracellular infection within the leaf tissues, thereby instigating plant pathogenesis. In *Phyllachora*, sexual ascomata are more commonly observed, whereas the asexual spermogonia are less conspicuous. They are often intermixed within the ascomata, which makes them difficult to observe. The sexual morphs are distinguished by conspicuous tar spots on the surface or underside of leaf tissues, typically appearing as oval, fusiform, or irregular shapes, surrounded by yellow necrotic lesions [1]. Morphologically, the sexual morph is characterized by a globose perithecium; numerous, branched paraphyses that are slightly longer than asci; asci that are eight-spored, persistent, cylindrical to fusiform, short-pedicellate, and often have an apical ring; and ascospores that are 1–3 seriate, fusiform to narrowly oval, hyaline, and sometimes have a gelatinous sheath [3,8,9]. The asexual manifestations of *Phyllachoraceae* have been reported as a coelomycetous morph, demonstrating either spermatic or disseminative properties [10,11,12]. *Phyllachora* species predominantly infect the leaves, stems, and bracts of host plants, causing tissue necrosis or large-scale dieback through subepidermal to intracellular infections [13,14,15,16].

*Poaceae* is an important grassland plant resource. In a special investigation of diseases on grassland plants, we collected specimens with obvious tar spots from *Pocaeae*. In the present study, three newly discovered species of *Phyllachora* found in China are introduced, supported by morphological and phylogenetic analyses. In addition, in reviewing the original references of *P. chloridis*, *P. graminis*, and *P. miscanthi*, these were found to be relatively old and in Chinese or abbreviated. Plus, the illustrations were simple. In molecular identification, the ITS sequence is short, while the ITS, LSU, and SSU are incomplete. Therefore, three other known species of *Phyllachora* are redescribed with detailed descriptions and illustrations. *Phyllachora* represent potential threats to forage health, with untreated infestations posing risks to grassland management and safety. Therefore, further investigation of these *Phyllachora* species holds promise for improving the future safety and management of grasslands.

## 2. Materials and Methods

### 2.1. Specimen Collection and Herbarium Deposit

Diseased leaf tissues from different hosts were collected from *Poaceae* plants on grasslands in Chengdu and the Liangshan Yi Autonomous Prefecture, Sichuan Province, China, between August 2022 and June 2023. Each fresh sample was meticulously placed in a self-sealing bag and then transported to the laboratory for further analysis. All specimens were stored in a −20 °C ultra-low temperature refrigerator at the Herbarium of Sichuan Agricultural University, Chengdu, China (SICAU).

### 2.2. Morphological Studies

The ascomata were examined with a dissecting microscope NVT-GG (Shanghai Advanced Photoelectric Technology Co., Ltd., Shanghai, China) fitted with a VS-800C micro-digital camera (Shenzhen Weishen Times Technology Co., Ltd., Shenzhen, China). Further microscopic analysis of the ascomata, peridium, paraphyses, asci, and ascospores, among others, was performed using an BX53 compound microscope equipped (Olympus Corporation, Japan) with an SD1600AC digital camera in conjunction with CapStudio (version 3.8.10.0) software from Image Technology Company, Suzhou, China. Subsequently, the iodine reaction of the ascus wall was tested in Melzer’s reagent. A minimum of 20 measurements was taken for each feature using Tarosoft^®^ Image Framework (version 0.9.7) software, developed by Tarosoft (R) in Nonthaburi, Thailand. The images were processed using Adobe Photoshop CC version 2022 software (Adobe Systems, San Jose, CA, USA). In addition, single ascospore isolations were performed according to the method described by Chomnunti et al. [17]. However, no spores had germinated.

### 2.3. DNA Extraction, Amplification, and Sequencing

The New Plant Genomic DNA Kit (Beijing Aidlab Biotechnologies Co., Ltd., Beijing, China) was used for total genomic DNA extraction from a single ascomata according to the manufacturer’s instructions. Amplifications of the ITS, LSU, and SSU gene fragments utilized three different primer pairs: ITS4/ITS5 for ITS [18], LROR/LR5 for LSU [19], and NS1/NS4 for SSU [18]. PCR reactions were performed according to the protocols provided by Golden Mix (Beijing TsingKe Biotech Co., Ltd., Beijing, China), including an initial denaturation at 98 °C for 2 min, followed by 30 cycles of denaturation at 98 °C for 10 s, annealing at 56 °C for 10 s, and extension at 72 °C for 10 s (for ITS and SSU) or 20 s (for LSU), with a final extension at 72 °C for 1 min. All PCR products were checked by electrophoresis in 2% agarose gels and sequenced at Hangzhou Youkang Biotech Co., Ltd., Chengdu, China, using forward and reverse primers.

### 2.4. Sequence Alignment and Phylogenetic Analyses

DNA sequences were aligned using Editseq 7.0.5.3 [20] to obtain consensus sequences. Sequences of *Phyllachoraceae* plant species for the establishment of multigene datasets were downloaded from GenBank (Table 1). The initial alignment of the sequences from individual loci was conducted using the MAFFT version 7 online service (https://mafft.cbrc.jp/alignment/server/, accession date: 1 April 2024), and then it was manually adjusted in BioEdit 7.0.5.3.

The phylogenetic relationships among the taxa were inferred using both maximum likelihood (ML) and Bayesian inference (BI) methods within Phylosuite software, version 1.2.3. [21]. *Telimena bicincta* (MM-108 and MM-133) was chosen as the outgroups. Maximum likelihood phylogenies were inferred using IQ-TREE [22] with an edge-linked partition model for 10,000 ultrafast bootstraps [22]. ModelFinder [23] was employed to select the best-fit partition model (edge-linked) based on the BIC criterion. The best-fit model according to BIC was TIM2e+I+G4 for ITS, LSU, and SSU. The Bayesian inference phylogenies were inferred using MrBayes under the partition model (2 parallel runs, 2,000,000 generations), within the initial 25% of the sampled data discarded as burn-in, and the best nucleotide substitution model for each locus was identified using ModelFinder of Phylosuite [21]. The best-fit model according to AIC was SYM+FQ+G4 for ITS, GTR+F+G4 for LSU, and TN+F+G4 for SSU. The resulting trees were visualized using FigTree v.1.4.3 [21], which is available at http://tree.bio.ed.ac.uk/software/figtree (accessed on 10 April 2024), and they were further refined in Adobe Illustrator CS6 2023 (v.27.6.0).

## 3. Results

### 3.1. Phylogenetic Analysis

Based on the ITS, LSU, and SSU sequence data, the molecular phylogenetic relationships were analyzed using six genera (*Ascovaginospora*, *Camarotella*, *Coccodiella*, *Neophyllachora*, *Phyllachora*, and *Polystigma*) within the *Phyllachoraceae*. Concatenated sequences from the three genes were obtained from 80 strains of *Phyllachoraceae*, resulting in a dataset with 4343 characters (LSU = 1176, ITS = 1237, SSU = 1929), including gaps. The best-scoring maximum likelihood consensus tree (lnL = −35,249.614) is depicted in Figure 1.

The SICAU 24-0044, SICAU 24-0045, SICAU 24-0048, SICAU 24-0049, SICAU 24-0046, SICAU 24-0047 are clustered within the genus *Phyllachora*, and since the ML and BI phylogenetic trees exhibited similar topologies, only the ML tree (Figure 1) is presented. Three new species were identified as *P. chongzhouensis*, *P. neidongensis*, and *P. huiliensis*. The three-gene phylogenetic analysis (Figure 1) suggests that the collections of *P. chongzhouensis* (SICAU 24-0044 and SICAU 24-0045) cluster together in a distinct clade, a sister to *P. thysanolaenae* (MFLU 16-2071), with MLBS/BYPP values of 83%/0.71. Additionally, the species *P. neidongensis* (SICAU 24-0046 and SICAU 24-0047) is a sister to *P. keralensis* (MHYAU:20083 and MHYAU:20082), with MLBS/BYPP values of 100%/0.99. Meanwhile, *P. huiliensis* (SICAU 24-0048 and SICAU 24-0049) formed a separate branch, which is a sister to *P. neidongensis* (SICAU 24-0046 and SICAU 24-0047), with MLBS/BYPP values of 93%/0.80. In addition, SICAU 24-0050 clustered with the strains of *P. miscanthi*, showing 100% MLBS and 1.00 BYPP. SICAU 24-0051 and SICAU 24-0052, clustered with the strains of *P. graminis*, with 78% MLBS. Collections SICAU 24-0053 and SICAU 24-0054, while different, also clustered with the strains of *P. chloridis*, showing 96% MLBS and 0.73 BYPP.

### 3.2. Taxonomy

*Phyllachora chloridis* Dayar., R.G. Shivas & K.D. Hyde. Mycosphere, 8(10): 1606 (2017). Figure 2.

=*Phyllachora chloridis-virgatae* J.C. Li, H.X. Wu, Y.Y. Li, X.H. Hao, J.Y. Song, Suwannarach & Wijayawardene, *Journal of Fungi* 8(5): 520 (2022).

Index Fungorum number: IF552804.

Description: Parasite associated with leaves of *Chloris virgata* (Poaceae), causing tar spot on leaves. Tar spots: 1.4–2.7 × 0.4–0.8 mm ($\bar{x}$ = 2.0 × 0.6 mm, *n* = 30) on both sides of the leaf, black, carbonaceous, fusiform, or cymbiform spots, scattered and shiny. Sometimes, there is a pale-yellow-to-yellow stripe at the edge of the tar spot. Sexual morph: *Ascomata:* 109–264 × 202–354 μm ($\bar{x}$ = 178 × 288 μm, *n* = 30), embedded within the leaf tissue, occupying the entire thickness of the leaf, often developing adjacent to neighboring ascomata and constrained by host vascular tissue, suboblate to subglobose, sometimes irregular, without obvious ostioles. *Peridium:* 18–40 μm wide, approximately 6–10 layers, dark brown to black; the darker cells are the outer layer, and the large, slightly paler cells are the inner layer. *Paraphyses:* 1.6–2.7 μm wide, numerous, persistent, filiform, unbranched, septate, slightly longer than the asci. *Asci:* 81–120 × 8–15 μm ($\bar{x}$ = 99 × 11 μm, *n* = 50), eight-spored, long, cylindrical, apex obtuse to rounded, negative staining with Melzer’s reagent. *Ascospores:* 11–18 × 5–9 μm ($\bar{x}$ = 14 × 7 μm, *n* = 50), uniseriate, sometimes overlapping, and angled; the cells are hyaline, ellipsoidal, and occasionally ovoid, one-celled, with some containing one or two large fat globules at the center. Asexual morph: Not observed.

Material examined: China, Sichuan Province, Liangshan Yi Autonomous Prefecture, Huili City, Xinfa Township (26.17′39.95″ N, 102.17′4.03″ E, alt. 1879 m), on leaves of *Chloris virgata* Sw., 28 September 2022, Qirong Sun & Chunlin Yang, SQR202209009 (SICAU 24-0053). GenBank accession numbers: ITS = PP785321, LSU = PP785310, SSU = PP785299; ibid. SQR2022090099 (SICAU 24-0054), GenBank accession numbers: ITS = PP785311, LSU = PP785322, SSU = PP785300.

Notes: According to the phylogenetic analysis, SICAU 24-0053 and SICAU 24-0054 are closely related to *Phyllachora chloridis*, with strong statistical support (100% MLBS, 1.00 BYPP). *P. chloridis* and *P. chloridis-virgatae* have the same host, and there was no difference in their molecular comparison [2,24]; so, they are considered to be the same species. Specimens SICAU 24-0053 and SICAU 24-0054, compared to the *P. chloridis* holotype (MFLU 15-0173), showed no differences in the ITS, LSU, and SSU sequences (all 100% identical) and matched the morphological description provided by Dayarathne et al. [2]. In conclusion, the collections SICAU 24-0053 and SICAU 24-0054 were classified within *P. chloridis*.

*Phyllachora chongzhouensis* Q.R. Sun, X.L. Xu & C.L. Yang, sp. nov. Figure 3.

Index Fungorum number: IF902115.

Etymology: Refer to the collection site, Chongzhou City, Sichuan Province, China.

Holotype: SICAU 24-0044

Description: Parasite on the leaves of *Phragmites australis* (Poaceae), causing tar spots on leaves. Tar spots: 1.2–2.8 × 0.3–0.7 mm ($\bar{x}$ = 1.7 × 0.5 mm, *n* = 30) on leaf surfaces, black, carbonaceous, elliptical to irregular shapes, often protruding above the leaf surface in a domed manner, and leaves are infested on both sides, solitary to gregarious. Sexual morph: *Ascomata:* 82–199 × 129–306 μm ($\bar{x}$ = 116 × 188 μm, *n* = 30), spindle-shaped, aligned, immersed in the leaf tissue, occupying the entire leaf thickness, and often developed next to the adjacent ascomata, confined by the host’s vascular tissue. *Peridium:* 16–27 μm wide, around 8–15 layers thick, from brown to black, containing host epidermal cells and sporadically encompassing the cuticle and inner layers. *Paraphyses:* 2.7–5.1 μm wide, numerous, persistent, filiform, unbranched, septate, slightly longer than the asci. *Asci:* 86–142 × 14–29 μm ($\bar{x}$ = 111 × 19 μm, *n* = 50), eight-spored, cylindrical, apex obtuse to rounded, pedicellate at posterior end, walls uniform in thickness, negative staining with Melzer’s reagent. *Ascospores:* 13–28 × 9–15 μm ($\bar{x}$ = 21 × 12 μm, *n* = 50), uniseriate, sometimes overlapping and oblique, hyaline; some are ellipsoidal, occasionally ovoid, one-celled, and ripe with a central depression. Asexual morph: Not observed.

Material examined: China, Sichuan Province, Chongzhou City, Jiayu Yangma Wetland Park (31°6′26.87″ N, 103°44′32.96″ E, alt. 730 m), on leaves of *Phragmites australis* (Cav.) Trin. ex Steud., 30 April 2024, Qirong Sun, SQR202404002 (SICAU 24-0044, holotype). GenBank accession numbers: ITS = PP785323, LSU = PP785312, SSU = PP785301; ibid SQR2024040022 (SICAU 24-0045), GenBank accession numbers: ITS = PP785324, LSU = PP785313, SSU = PP785302.

Notes: Multi-locus phylogenetic analyses utilizing a concatenated ITS, LSU, and SSU sequence dataset revealed that the new species *P. chongzhouensis* is related to *P. thysanolaenae* in a subclade with 83% MLBS and 0.71 BYPP statistical support (Figure 1). However, *P. chongzhouensis* (SICAU 24-0044) exhibits distinctly different morphological characteristics compared to *P. thysanolaenae* (MFLU 16-2071). Specifically, *P. chongzhouensis* features larger asci (86–142 × 14–29 μm vs. 100–126 × 13–18 μm) and wider asci walls (9–15 μm vs. 4–5 μm). Moreover, the ascospores of *P. chongzhouensis* are ellipsoidal with a central depression, whereas the ascospores of *P. thysanolaenae* (MFLU 16-2071) are cylindrical–fusiform and surrounded by a mucilaginous sheath [25]. In the comparison of the SSU sequences, there is a 7.37% nucleotide difference between *P. chongzhouensis* (SICAU 24-0044) and its phylogenetically affiliated *P. thysanolaenae* (MFLU 16-2071). Additionally, there are no available ITS and LSU sequences for comparison. These findings on the morphological features and molecular phylogenetic characteristics strongly support the proposal for establishing *P. chongzhouensis* as a new species, as recommended by Hyde et al. [26].

*Phyllachora graminis* (Pers.) Fuckel, Jahrb. Nassauischen Vereins Naturk. 23-24: 216 (1870), Figure 4.

Index Fungorum number: IF200927.

Description: Parasite on leaves of *Lolium perenne* (Poaceae), causing tar spots on leaves. Tar spots: 0.9–1.9 × 0.3–0.6 mm ($\bar{x}$ = 1.3 × 0.4 mm, *n* = 20) on both sides of the leaf, black, carbonaceous, elliptical to irregular, solitary to gregarious, with a halo around the periphery. Sexual morph: *Ascomata:* 102–299 × 109–273 μm ($\bar{x}$ = 196 × 172 μm, *n* = 20), immersed within the leaf tissue, occupying the entire leaf thickness, often developing adjacent to neighboring ascomata and confined by host vascular tissue, ellipsoid to spherical, sometimes irregular. *Peridium:* 7–17 μm wide, approximately 8–10 layers, dark brown to black; the darker cells are the outer layer, and the large, slightly paler cells are the thin-walled inner layer. *Paraphyses:* 1.2–1.9 μm wide, numerous, persistent, filiform, unbranched, septate, slightly longer than asci. *Asci:* 64–101 × 6–10 μm ($\bar{x}$ = 85 × 8 μm, *n* = 50), eight-spored, long, cylindrical, slightly curved, apex obtuse to rounded, negative staining with Melzer’s reagent. *Ascospores:* 8–15 × 4–7 μm ($\bar{x}$ = 11 × 6 μm, *n* = 50), uniseriate, sometimes overlapping and oblique, hyaline, ellipsoidal, one-celled, some with fat globules in the center, some middle depressions. Asexual morph: Not observed.

Material examined: China, Sichuan Province, Chengdu City, Qionglai Jiguan Townships (30°17′2.13″ N 103°15′48.56″ E, alt. 551 m), on leaves of *Lolium perenne* Linn., 23 May 2023, Qirong Sun & Liping Gao, SQR202305038 (SICAU 24-0051). GenBank accession numbers: ITS = PP785317, LSU = PP785306, SSU = PP785295; China, Sichuan Province, Chengdu City, Jinjiang District, (30°33′53.46″ N, 104°9′10.08″ E, alt. 755 m), on leaves of *Lolium perenne* Linn., 5 June 2023, Qirong Sun & Liping Gao, SQR202305052 (SICAU 24-0052). GenBank accession numbers: ITS = PP785318, LSU = PP785307, SSU = PP785296.

Notes: The collections SICAU 24-0051 and SICAU 24-0052 clustered together with the known species *Phyllachora graminis* with a 78% ML bootstrap support value (Figure 1). Nucleotide comparisons of SICAU 24-0051 showed high homology with the sequences of *P. graminis* (RoKi3084). In the LSU and SSU regions, the similarities are 99.82% (549/550, 1 gap) and 100% (420/420, 0 gaps), respectively. And the same similarities were observed in SICAU 24-0052. Furthermore, morphological analysis of the new collection aligns with the description of *P. graminis* provided by Orton [27]. Based on comprehensive evidence, SICAU 24-0051 and SICAU 24-0052 can be classified within *P. graminis*.

*Phyllachora huiliensis* Q.R. Sun & C.L. Yang, sp. nov., Figure 5.

Index Fungorum number: IF902117.

Etymology: Refer to the collection site: Huili City, Sichuan Province, China.

Holotype: SICAU 24-0048.

Description: Parasite on leaves of *Bothriochloa ischaemum* (Poaceae), causing tar spots on leaves. Tar spots: 0.1–2.2 × 0.2–0.6 mm ($\bar{x}$ = 1.3 × 0.4 mm, *n* = 20) on the upper leaf surface, fusiform or cymbiform, amphigenous, solitary to gregarious, sometimes with orifices. Sexual morph: *Ascomata:* 86–175 × 132–224 μm ($\bar{x}$ = 123 × 171 μm, *n* = 30), immersed in the leaf tissue, occupying the entire leaf thickness, often developing next to the adjacent ascomata and confined by the host vascular tissue, suboblate to subglobose, sometimes irregular. *Peridium:* 17–27 μm wide, approximately 6–10 layers, dark brown; the darker cells are the outer layer, and the large, slightly paler cells are the thin-walled inner layer. *Paraphyses:* 1.8–3.1 μm wide, numerous, persistent, filiform, unbranched, septate, slightly longer than asci. *Asci:* 74–112 × 7–12 μm ($\bar{x}$ = 92 × 9 μm, *n* = 50), eight-spored, long, cylindrical, apex obtuse to rounded, negative staining with Melzer’s reagent. *Ascospores:* 11–17 × 5–8 μm ($\bar{x}$ = 13 × 6 μm, *n* = 50), uniseriate, occasionally overlapping, hyaline, and ellipsoidal or ovoid cells, some of which contain one or two large lipid droplets centrally. Asexual morph: Not observed.

Material examined: China, Sichuan Province, Liangshan Yi Autonomous Prefecture, Huili City, Neidong Township (26°35′5.99″ N 102°20′50.30″ E, alt. 2143 m), on leaves of *Bothriochloa ischaemum* (L.) Keng, 24 July 2023, Qirong Sun & Chunlin Yang, SQR202307002 (SICAU 24-0048, holotype). GenBank accession numbers: ITS = PP785319, LSU = PP785308, SSU = PP785297; ibid SQR2023070022 (SICAU 24-0049), GenBank accession numbers: ITS = PP785320, LSU = PP785309, SSU = PP785298.

Notes: In the phylogenetic tree, the collections of *Phyllachora huiliensis* cluster in an independent subclade within *Phyllachora* (Figure 1). Morphologically, *P. huiliensis* (SICAU 24-0048, holotype) displays distinct characteristics compared to *P. neidongensis* (SICAU 24-0046, holotype). *P. huiliensis* has smaller ascomata (86–175 × 132–224 μm vs. 145–481 × 196–480 μm) than the latter. Furthermore, the paraphyses of *P. neidongensis* are branched, and they have gelatinous sheaths around the ascospores, which is not observed in *P. huiliensis*. Nucleotide comparisons reveal notable differences between *P. huiliensis* (SICAU 24-0048, holotype) and *P. neidongensis* (SICAU 24-0046, holotype). The nucleotide differences are 19.46% (86/443, 0 gaps), 9.13% (68/745, 0 gaps), and 1.81% (17/937, 0 gaps) in the ITS, LSU, and SSU regions, respectively. Hence, we describe *P. huiliensis* as a new species in *Phyllachora*, as recommended by Maharachchimbura et al. [3].

*Phyllachora miscanthi* Syd. & P. Syd., Annales Mycologici 15 (3-4): 227 (1917), Figure 6.

Index Fungorum number: IF165328.

Description: Parasite on leaves of *Miscanthus floridulus* (Poaceae), causing tar spot on leaves. Tar spots: 1.4–2.7 × 0.4–0.8 mm ($\bar{x}$ = 2.0 × 0.6 mm, *n* = 30) on both sides of the leaf surface, black, carbonaceous, ellipse to irregularity, amphigenous, solitary to gregarious. Sexual morph: *Ascomata:* 195–614 × 178–399 μm ($\bar{x}$ = 178 × 288 μm, *n* = 20), immersed in the leaf tissue, occupying the entire leaf thickness, often developing next to adjacent ascomata and confined by host vascular tissue, irregular in shape, and no obvious ostiolate. *Peridium:* 7–18 μm wide, approximately 10–13 layers, dark brown to black; the darker cells are the outer layer, and the large, slightly paler cells are the thin-walled inner layer. *Paraphyses:* 1.7–2.5 μm wide, numerous, persistent, filiform, unbranched, septate, slightly longer than the asci. *Asci:* 83–135 × 14–27 μm ($\bar{x}$ = 106 × 20 μm, *n* = 50), eight-spored, long, cylindrical, sometimes wider in the middle, blunt to rounded at the apex, with a stalk at the base, negative staining with Melzer’s reagent. *Ascospores:* 19–29 × 7–11 μm ($\bar{x}$ = 23 × 9 μm, *n* = 50), large, uniseriate, elliptical, with the other end more pointed, single or double arrangement. Asexual morph: Not observed.

Material examined: China, Sichuan Province, Chengdu City, Qionglai City, Haihong Community (30°17′2.13″ N, 103°15′48.56″ E, alt. 551 m), on leaves of *Miscanthus floridulus* (Lab.) Warb. ex Schum. et Laut., 23 May 2023, Qirong Sun & Liping Gao, SQR202305037 (SICAU 24-0050). GenBank accession numbers: ITS = PP785316, LSU = PP785305, SSU = PP785294.

Notes: Phylogenetic analyses revealed that SICAU 24-0050 forms a sub-branch within *Phyllachora miscanthi* (99% MLBS and 1.00 BYPP) (Figure 1), specifically close to *P. miscanthi* (MHYAU:167). Sequence comparisons demonstrated a high similarity in their LSU (99.6%, 455/457, 2 gaps) and SSU (99.6%, 790/793, 3 gaps). Furthermore, the morphological analysis of the new collection aligns with the description of *P. miscanthi* provided by Zhang Z. Y. et al. [28]. Based on comprehensive evidence, SICAU 24-0050 can be classified within *P. miscanthi.*

*Phyllachora neidongensis* Q.R. Sun & C.L. Yang, sp. nov., Figure 7.

Index Fungorum number: IF902116.

Etymology: Refer to the collection site, Neidong Township, Huili City, Sichuan Province, China.

Holotype: SICAU 24-0046.

Description: Parasite on leaves of *Themeda triandra* (Poaceae), causing tar spots on leaves. Tar spots: 1.1–2.5 × 0.4–1.0 mm ($\bar{x}$ = 1.6 × 0.6 mm, *n* = 30) on the upper leaf surface, fusiform or cymbiform, solitary to gregarious, black, carbonaceous, with a yellow halo of discolored host tissue. Sexual morph: *Ascomata:* 145–481 × 196–480 μm ($\bar{x}$ = 327 × 346 μm, *n* = 30), immersed within the leaf tissue, occupying the entire thickness, often developing adjacent to the neighboring ascomata and confined by the host vascular tissue; the structures are suboblate to subglobose, occasionally irregular in shape, and lack obvious ostioles. *Peridium:* 14–35 μm wide, approximately 6–8 layers, dark brown; the darker cells are the outer layer, and the large, slightly paler cells are the thin-walled inner layer. *Paraphyses:* 1.1–2.8 μm wide, numerous, persistent, filiform, branched, septate, slightly longer than the asci. *Asci:* 77–135 × 8–14 μm ($\bar{x}$ = 110 × 11 μm, *n* = 50), eight-spored, long, cylindrical, pedicellate at the posterior end, walls uniform in thickness, negative staining with Melzer’s reagent. *Ascospores:* 12–19 × 7–10 μm ($\bar{x}$ = 16 × 8 μm, *n* = 50), uniseriate, with cells occasionally overlapping and oblique; the hyaline, ellipsoidal-to-ovoid cells consist of a single cell type, some of which contain one or two large lipid droplets centrally located within the cell, all surrounded by a gelatinous sheath. Asexual morph: Not observed.

Material examined: China, Sichuan Province, Liangshan Yi Autonomous Prefecture, Huili City, Neidong Township (26°35′46.14″ N 102°20′41.96″ E, alt. 2093 m), on leaves of *Themeda triandra* Forsk., 26 September 2022, Qirong Sun & Chunlin Yang. SQR202209010 (SICAU 24-0046, holotype). GenBank accession numbers: ITS = PP785325, LSU = PP785314, SSU = PP785303; ibid SQR2022090100 (SICAU 24-0047), GenBank accession numbers: ITS = PP785326, LSU = PP785315, SSU = PP785304.

Notes: The three-gene phylogenetic analyses show that *Phyllachora neidongensis* is related to *P. keralensis* (99%MLBS, 1.00 BYPP). Microscopically, *P. neidongensis* exhibits larger ascomata (145–481 × 196–480 μm vs. 71–96 × 74–165 μm), longer asci (77–135 × 8–14 μm vs. 49–79 × 11–14 μm), and larger ascospores (12–19 × 7–10 μm vs. 9–13 × 6–7 μm) than *P. keralensis*, as described by Teng et al. [29]. Additionally, the paraphyses of *P. neidongensis* are branched, while those of the former are not. Nucleotide comparisons reveal significant differences between *P. neidongensis* (SICAU 24-0046) and *P. keralensis* (MHYAU:20083), viz. 23.04% (91/395, 0 gap), and 6.40% (49/499, 0 gap) in the ITS and LSU, respectively. Therefore, *P. neidongensis* strain SICAU 24-0046 was proposed as a new species.

## 4. Discussion

*Phyllachora* species are thermophilic, more likely to occur in hot summer conditions, and are mainly found in tropical and subtropical regions [12,30,31,32]. To date, 84 species of *Phyllachora* have been recognized in China, and they are mainly distributed in southern China [33,34,35,36], viz., Yunnan, Guangxi, Guangdong, and Sichuan Provinces. Yunnan Province is the most studied [37]. Only 22 species have been identified at the molecular level, and most of their host plants are *Poaceae*, which are most frequently found in high-temperature and high-humidity environments [1,2,9,24]. Thus, we inferred that the hot and humid summer climate and rich plant diversity in Sichuan Province facilitate the occurrence and infiltration of *Phyllachora* species. However, there are very few studies on *Phyllachora* fungi, with most focusing only on morphological identification, such as *P. cynodontis*, *P. graminis*, *P. lespedezae*, *P. quadraspora*, and *P. sacchari* [33,34,36]. These fungi infect more than 10 types of host plants, including *Lespedeza bicolor* (Fabaceae), *Cynodon dactylon*, *Lolium perenne*, *Phyllostachys sulphurea*, *Eleusine indica*, and *Miscanthus sinensis* (*Poaceae*) [33,34,36]. Until now, only one *Phyllachora heterocladae* species on *Phyllostachys heteroclada* has been identified with both morphological and molecular analyses [9]. Most *Phyllachora* species studies lack the support of comprehensive morphological and phylogenetic analyses.

In this study, the host plants of *Phyllachora* were *Bothriochloa ischaemum*, *Phragmites australis*, *Themeda triandra*, *Chloris virgata*, *Lolium perenne*, and *Miscanthus floridulus*, in which *B. ischaemum*, *P. australis*, and *T. triandra* were newly discovered. It is well known that *Phyllachora* species are often recognized as obligate fungi, but *P. virgatae* and *P. jiaensis* have been reported from the same host (*Chloris virgata*) on the grass resources of China [1]. Similarly, *P. miscanthi* and *P. graminis* have been found on *Lolium perenne* in Chengdu and Ya’an City, Sichuan Province [33]. Therefore, *Phyllachora* species cannot be easily distinguished based on different host plants, and detailed morphological characteristics and molecular analyses are essential to reveal new taxa and enhance understanding of their diversity. 

In Sichuan Province, China, the vast plateau grasslands, which harbor diverse ecosystems and rich forage species, have esthetic, ecological, and economic value. However, the persistence of irrational management practices and the effects of climate change have led to periodic, widespread outbreaks of grassland diseases. Species of the genus *Phyllachora*, in particular, are emerging as major pathogens affecting grassland plant diseases [14,15]. During our investigation, a large area of *Phragmites australis* in Yangma Wetland Park, Chongzhou City, was infested with tar spot, characterized by dense black spots on yellow, withered, and dead leaves. This has also been observed in other wetland parks in Chengdu. In addition, the incidence of tar spot on *Chloris virgata*, located in Huili City, was also serious in this survey. In previous studies, Liu et al. [36] proposed that the species in genus *Phyllachora* are important plant pathogenic fungi, which are extremely harmful to herbages, as exemplified by *Phyllachora maydis* in the United States, causing serious effects on the quality and yield [13,14,15,16]. Despite these challenges, research on fungal diseases remains scarce. Therefore, comprehensive research is urgently needed to trace the disease type on grasslands and to protect their ecological security.

## Figures and Tables

**Figure 1 jof-10-00588-f001:**
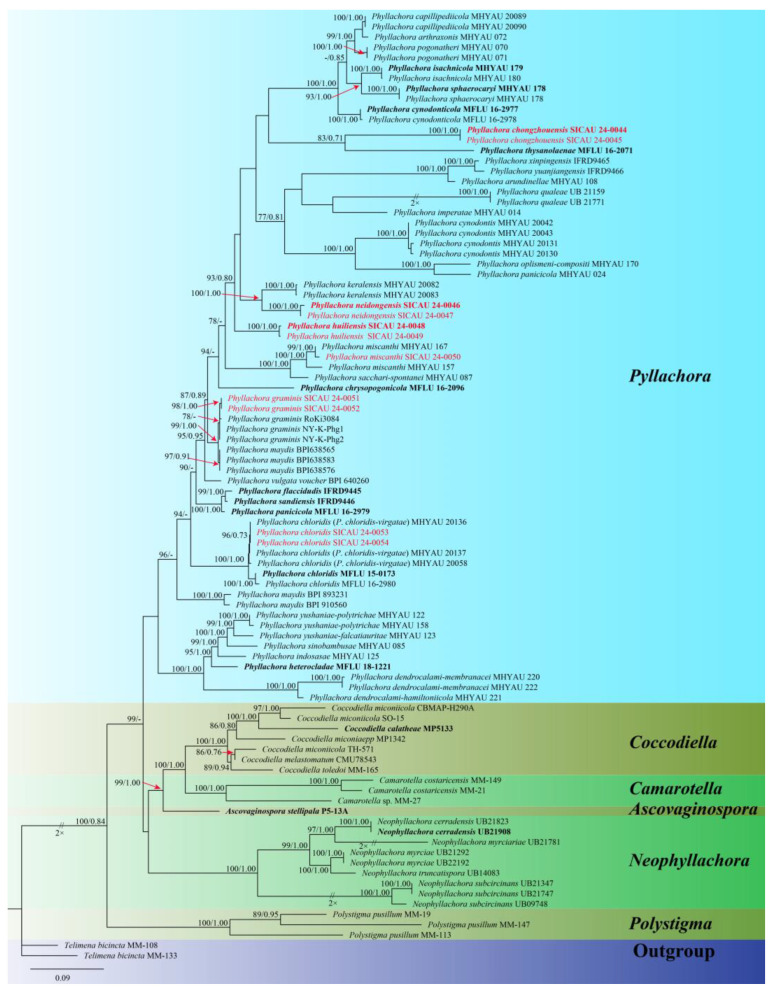
Phylogenetic tree generated from maximum likelihood analysis using the concatenated sequences of the ITS, LSU, and SSU loci of the genera in *Phyllachoracea*. Notes are marked with maximum likelihood bootstrap proportions ≥ 70% (**left**) and Bayesian inference posterior probability values ≥ 0.7 (**right**) (MLBP/BIPP). Some inter-section support is marked with red arrows, ex-type or ex-epitype strains are highlighted in bold, and study species are indicated in red.

**Figure 2 jof-10-00588-f002:**
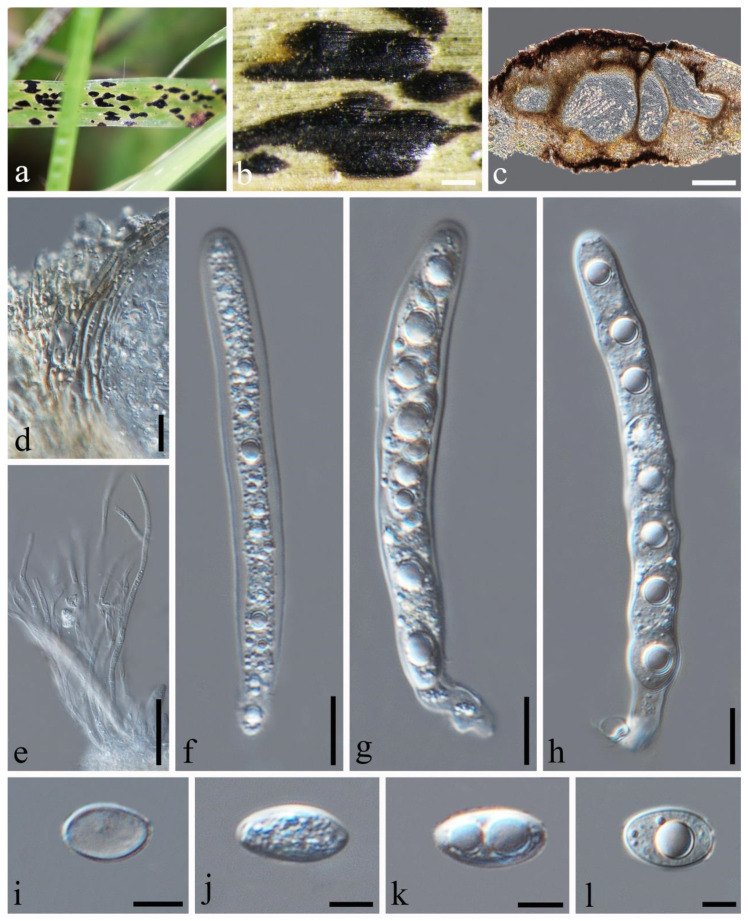
*Phyllachora chloridis* (SICAU 24-0053). (**a**,**b**) Black spots on leaves. (**c**) Vertical cross-section of ascoma. (**d**) Peridium. (**e**) Paraphyses. (**f**–**h**) Asci. (**i**–**l**) Ascospores. Scale bars: 300 μm (**b**), 100 μm (**c**), 10 μm (**d**–**h**), 5 μm (**i**–**l**).

**Figure 3 jof-10-00588-f003:**
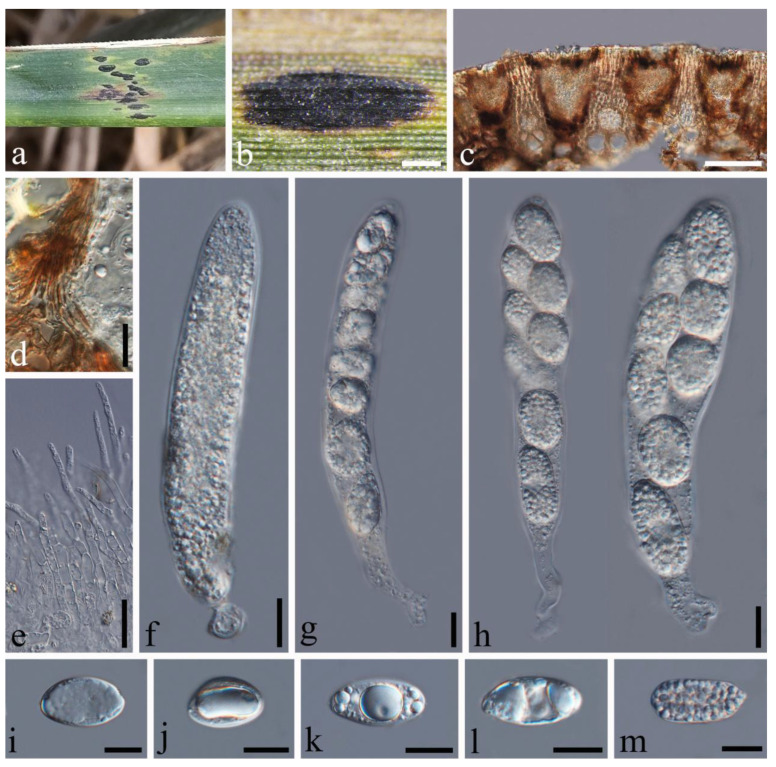
*Phyllachora chongzhouensis* (SICAU 24-0044, holotype). (**a**,**b**) Black spots on leaves. (**c**) Vertical cross-section of ascoma. (**d**) Peridium. (**e**) Paraphyses. (**f**–**h**) Asci. (**i**–**m**) Ascospores. Scale bars: 400 μm (**b**), 50 μm (**c**), 10 μm (**d**–**m**).

**Figure 4 jof-10-00588-f004:**
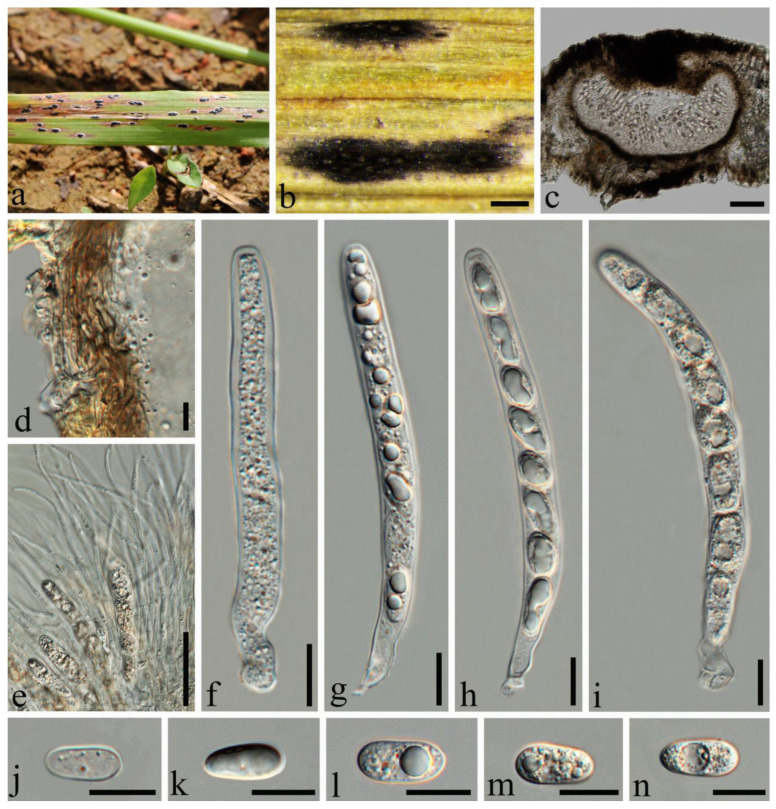
*Phyllachora graminis* (SICAU 24-0051). (**a**,**b**) Black spots on leaves. (**c**) Vertical cross-section of ascoma. (**d**) Peridium. (**e**) Paraphyses. (**f**–**i**) Asci. (**j**–**n**) Ascospores. Scale bars: 300 μm (**b**), 50 μm (**c**), 10 μm (**d**–**n**).

**Figure 5 jof-10-00588-f005:**
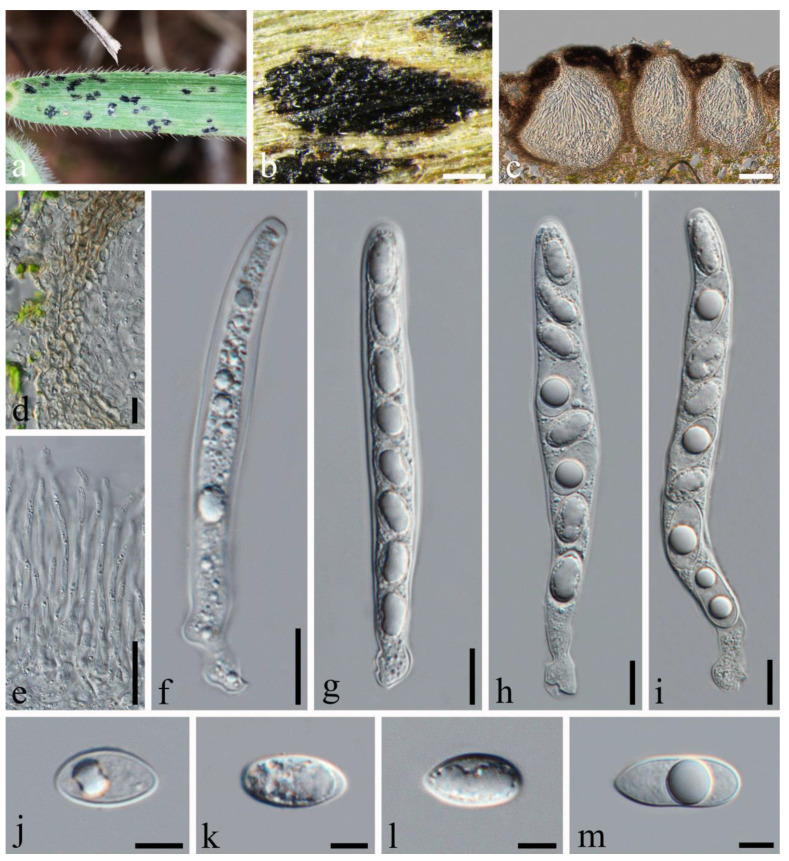
*Phyllachora huiliensis* (SICAU 24-0048, holotype). (**a**,**b**) Black spots on leaves. (**c**) Vertical cross-section with orifices. (**d**) Peridium. (**e**) Paraphyses. (**f**–**i**) Asci. (**j**–**m**) Ascospores. Scale bars: 400 μm (**b**), 50 μm (**c**), 10 μm (**d**–**i**), 5 μm (**j**–**m**).

**Figure 6 jof-10-00588-f006:**
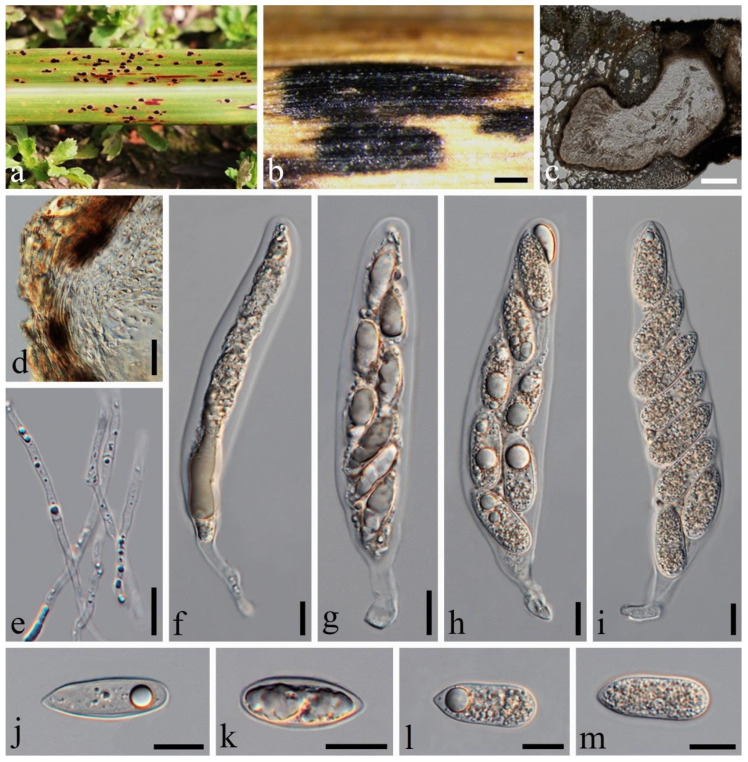
*Phyllachora miscanthi* (SICAU 24-0050). (**a**,**b**) Black spots on leaves. (**c**) Vertical cross-section of ascoma. (**d**) Peridium. (**e**) Paraphyses. (**f**–**i**) Asci. (**j**–**m**) Ascospores. Scale bars: 300 μm (**b**), 100 μm (**c**), 10 μm (**d**–**m**).

**Figure 7 jof-10-00588-f007:**
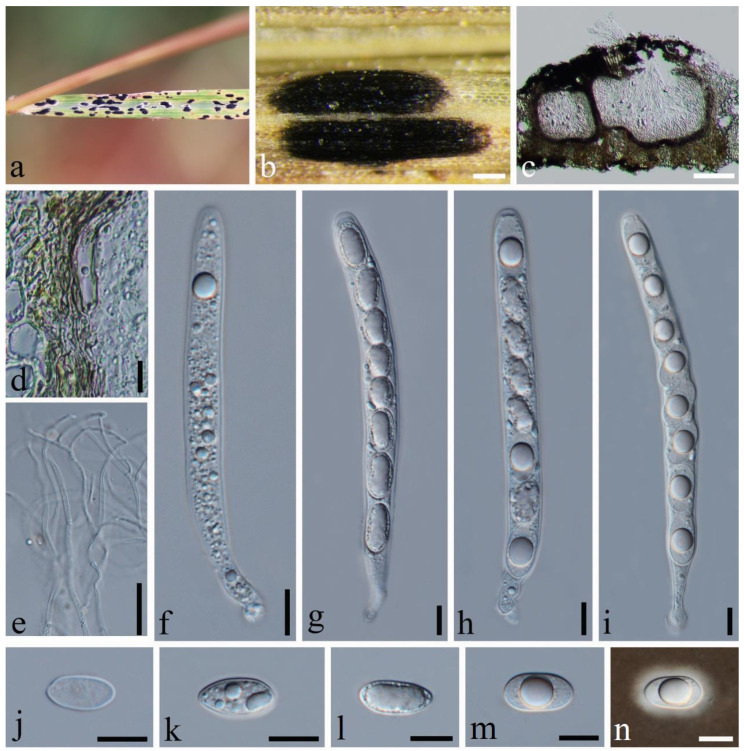
*Phyllachora neidongensis* (SICAU 24-0046, holotype). (**a**,**b**) Black spots on leaves. (**c**) Vertical cross-section of ascoma. (**d**) Peridium. (**e**) Paraphyses. (**f**–**i**) Asci. (**j**–**m**) Ascospores. (**n**) Ascospore with mucilaginous sheath. Scale bars: 200 μm (**b**), 100 μm (**c**), 10 μm (**d**–**n**).

**Table 1 jof-10-00588-t001:** Samples used for multigene phylogenetic analysis ^a^. GenBank Numbers ^b^.

Species	Location	Strain	Host (Family)	GenBank Accession Numbers		
				LSU	ITS	SSU
*Ascovaginospora stellipala* ^T^	Northern Wisconsin	P5-13A	*Carex limosa* (Cyperaceae)	U85088	-	U85087
*Camarotella costaricensis*	Panama	MM-21	*Acrocomia aculeata* (Arecaceae)	KX430490	KX451900	KX451851
*Camarotella costaricensis*	Panama	MM-149	*Acrocomia aculeata* (Arecaceae)	KX430484	KX451913	KX451863
*Camarotella amarotella* sp.	Panama	MM-27	Arecaceae	KX430492	KX451901	KX451852
*Coccodiella calatheae* ^T^	Panama	MP5133	*Calathea crotalifera* (Marantaceae)	MF460370	MF460366	MF460376
*Coccodiella melastomatum*	Venezuela	CMU78543	*Miconia* sp. (Melastomataceae)	-	-	U78543
*Coccodiella miconiae*	Northern Wisconsin	ppMP1342	Melastomataceae	KX430506	MF460365	KX451871
*Coccodiella miconiicola*	Panama	TH-571	*Ossaea micrantha* (Melastomataceae)	KX430512	-	KX451880
*Coccodiella miconiicola*	Panama	CBMAP-H290A	*Miconia* sp. (Melastomataceae)	MF460373	MF460368	MF460379
*Coccodiella miconiicola*	Ecuador	SO-15	*Graffenrieda* sp. (Melastomataceae)	MF460374	MF460369	MF460380
*Coccodiella toledoi*	Ecuador	MM-165	Melastomataceae	KX430488	KX451917	KX451865
*Neophyllachora cerradensis*	Brazil	UB21823	*Myrcia torta* (Myrtaceae)	-	KC683470	-
*Neophyllachora cerradensis* ^T^	Brazil)	UB21908	*Myrcia pinifolia* (Myrtaceae)	-	KC683471	-
*Neophyllachora myrciae*	Brazil	UB21292	*Myrcia pallens* (Myrtaceae)	-	KC683463	-
*Neophyllachora myrciae*	Brazil	UB22192	*Myrcia variabilis* (Myrtaceae)	-	KC683476	-
*Neophyllachora myrciariae* ^T^	Brazil	UB21781	*Myrciaria delicatula* (Myrtaceae)	-	KC683469	-
*Neophyllachora subcircinans*	Brazil	UB09748	*Psidium australe* (Myrtaceae)	-	KC683441	-
*Neophyllachora subcircinans*	Brazil Paraguay	UB21347	*Psidium guineense* (Myrtaceae)	-	KC683466	-
*Neophyllachora subcircinans*	Brazil Paraguay	UB21747	*Psidium australe* (Myrtaceae)	-	KC683467	KC902622
*Neophyllachora truncatispora*	Brazil	UB14083	*Myrcia camapuanensis* (Myrtaceae)	-	KC683448	KC902614
*Phyllachora arthraxonis*	China: Yunnan	MHYAU: 072	*Arthraxon hispidus* (Poaceae)	MG269803	MG269749	-
*Phyllachora arundinellae*	China: Yunnan	MHYAU: 108	*Arundinella setosa* (Poaceae)	MG269815	MG269761	-
*Phyllachora capillipediicola*	China: Yunnan	MHYAU 20089	Poaceae	MG356698	KY498084	-
*Phyllachora capillipediicola*	China: Yunnan	MHYAU: 20090	Poaceae	MG356699	KY498115	-
*Phyllachora chloridis* ^T^	Thailand	MFLU 15-0173	*Chloris* sp. (Poaceae)	MF197499	KY594026	MF197505
*Phyllachora chloridis*	Thailand	MFLU 16-2980	*Chloris* sp. (Poaceae)	MF197500	KY594027	MF197506
*Phyllachora chloridis* (*P. chloridis-virgatae*)	China: Yunnan	MHYAU 20136	*Chloris virgata* (Poaceae)	MG356685	KY498122	-
*Phyllachora chloridis* (*P. chloridis-virgatae*)	China: Yunnan	MHYAU: 20058	*Chloris virgata* (Poaceae)	MG356683	KY498102	-
*Phyllachora chloridis* (*P. chloridis-virgatae*)	China: Yunnan	MHYAU 20137	*Chloris virgata* (Poaceae)	MG356686	KY498092	-
** *Phyllachora chloridis* **	**China: Sichuan**	**SICAU 24-0053**	***Chloris virgata* (Poaceae) **	**PP785310**	**PP785321**	**PP785299**
** *Phyllachora chloridis* **	**China: Sichuan**	**SICAU 24-0054**	***Chloris virgata* (Poaceae) **	**PP785311**	**PP785322**	**PP785300**
*Phyllachora chrysopogonicola* ^T^	Thailand	MFLU 16-2096	*Chrysopogon zizanioides* (Poaceae)	MF372146	MF372145	-
*Phyllachora cynodonticola* ^T^	Thailand	MFLU 16-2977	*Cynodon* sp. (Poaceae)	MF197501	KY594024	MF197507
*Phyllachora cynodonticola*	Thailand	MFLU 16-2978	*Imperata* sp. (Poaceae)	MF197502	KY594025	MF197508
*Phyllachora cynodontis*	China: Yunnan	MHYAU 20042	*Cynodon dactylon* (Poaceae)	KY498080	KY471328	-
*Phyllachora cynodontis*	China: Yunnan	MHYAU: 20043	*Cynodon dactylon* (Poaceae)	KY498081	KY471329	-
*Phyllachora cynodontis*	China: Yunnan	MHYAU 20131	*Cynodon dactylon* (Poaceae)	KY498079	KY471327	-
*Phyllachora cynodontis*	China: Yunnan	MHYAU 20130	*Cynodon dactylon* (Poaceae)	KY498083	KY471331	-
** *Phyllachora chongzhouensis* **	**China: Sichuan**	**SICAU 24-0044**	***Phragmites australis* (Poaceae) **	**PP785312**	**PP785323**	**PP785301**
** *Phyllachora chongzhouensis* **	**China: Sichuan**	**SICAU 24-0045**	***Phragmites australis* (Poaceae) **	**PP785313**	**PP785324**	**PP785302**
*Phyllachora dendrocalami-hamiltoniicola*	China: Yunnan	MHYAU 221	*Dendrocalamus hamiltonii* (Poaceae)	MK614118	-	-
*Phyllachora dendrocalami-membranacei*	China: Yunnan	MHYAU 220	*Dendrocalamus membranaceus* (Poaceae)	MK614117	MK614102	-
*Phyllachora dendrocalami-membranacei*	China: Yunnan	MHYAU 222	*Dendrocalamus membranaceus* (Poaceae)	MK614119	MK614103	-
*Phyllachora flaccidudis*	China	IFRD9445	*Cenchrus flaccidus* (Poaceae)	ON072101	ON075524	ON072097
*Phyllachora graminis*	Germany	RoKi3084	*Arrhenatherum elatius* (Poaceae)	KX430507	-	KX451872
*Phyllachora graminis*	Germany	NY-K-Phg2	Poaceae	MW774239	-	-
*Phyllachora graminis*	Germany	NY-K-Phg1	Poaceae	MW774238	-	-
** *Phyllachora graminis* **	**China: Sichuan**	**SICAU 24-0051**	***Lolium perenne* (Poaceae) **	**PP785306**	**PP785317**	**PP785295**
** *Phyllachora graminis* **	**China: Sichuan**	**SICAU 24-0052**	***Lolium perenne* (Poaceae) **	**PP785307**	**PP785318**	**PP785296**
*Phyllachora heterocladae* ^T^	China: Sichuan	MFLU 18-1221	*Phyllostachys heteroclada* (Poaceae)	MK296472	MK305902	MK296468
** *Phyllachora huiliensis* **	**China: Sichuan**	**SICAU 24-0048**	***Bothriochloa ischaemum* (Poaceae) **	**PP785308**	**PP785319**	**PP785297**
** *Phyllachora huiliensis* **	**China: Sichuan**	**SICAU 24-0049**	***Bothriochloa ischaemum* (Poaceae) **	**PP785309**	**PP785320**	**PP785298**
*Phyllachora imperatae*	China: Yunnan	MHYAU: 014	*Imperata cylindrica* (Poaceae)	MG269800	MG269746	-
*Phyllachora indosasae*	China: Yunnan	MHYAU 125	*Indosasa hispida* (Poaceae)	MG195662	MG195637	-
*Phyllachora isachnicola* ^T^	China: Yunnan	MHYAU: 179	*Isachne albens* (Poaceae)	MH018563	MH018561	-
*Phyllachora isachnicola*	China: Yunnan	MHYAU: 180	*Isachne albens* (Poaceae)	MH018564	MH018562	-
*Phyllachora keralensis*	China: Yunnan	MHYAU: 20082	Poaceae	MG269792	KY498106	MH992447
*Phyllachora keralensis*	China: Yunnan	MHYAU: 20083	Poaceae	MG269793	KY498088	-
*Phyllachora maydis*	USA	BPI 893231	*Zea mays* (Poaceae)	-	KU184459	-
*Phyllachora maydis*	USA	BPI 910560	*Zea mays* (Poaceae)	-	MG881846	-
*Phyllachora maydis*	USA	BPI638583	*Zea mays* (Poaceae)	-	OL342922	-
*Phyllachora maydis*	USA	BPI638576	*Zea mays* (Poaceae)	-	OL342921	-
*Phyllachora maydis*	USA	BPI638565	Zea mays (Poaceae)	-	OL342920	-
*Phyllachora miscanthi*	China: Yunnan	MHYAU: 167	*Miscanthus sinensis* (Poaceae)	MG195669	MG195644	-
*Phyllachora miscanthi*	China: Yunnan	MHYAU: 157	*Miscanthus sinensis* (Poaceae)	MG195668	MG195643	-
** *Phyllachora miscanthi* **	**China: Sichuan**	**SICAU 24-0050**	***Miscanthus floridulus* (Poaceae) **	**PP785305**	**PP785316**	**PP785294**
** *Phyllachora neidongensis* **	**China: Sichuan**	**SICAU 24-0046**	***Themeda triandra* (Poaceae) **	**PP785314**	**PP785325**	**PP785303**
** *Phyllachora neidongensis* **	**China: Sichuan**	**SICAU 24-0047**	***Themeda triandra* (Poaceae) **	**PP785315**	**PP785326**	**PP785304**
*Phyllachora oplismeni-compositi*	China: Yunnan	MHYAU: 170	*Oplismenus compositus* (Poaceae)	MG195673	MG195648	-
*Phyllachora panicicola*	China: Yunnan	MHYAU: 024	*Panicum khasianum* (Poaceae)	MG195674	MG195649	-
*Phyllachora panicicola* ^T^	China	MFLU 16-2979	*Panicum* sp. (Poaceae)	MF197503	KY594028	MF197504
*Phyllachora pogonatheri*	China: Yunnan	MHYAU:071	*Pogonatherum paniceum* (Poaceae)	MG269802	MG269748	-
*Phyllachora pogonatheri*	China: Yunnan	MHYAU: 070	*Pogonatherum crinitum* (Poaceae)	MG269801	MG269747	-
*Phyllachora qualeae*	Unknown	UB 21159	*Qualea multiflora* (Vochysiaceae)	-	KU682781	-
*Phyllachora qualeae*	Unknown	UB 21771	*Qualea multiflora* (Vochysiaceae)	-	KU682780	-
*Phyllachora sandiensis* ^T^	China: Shaanxi	IFRD9446	*Cenchrus flaccidus* (Poaceae)	ON075528	ON075525	ON072098
*Phyllachora sacchari-spontanei*	China: Yunnan	MHYAU: 087	*Saccharum spontaneum* (Poaceae)	MG195670	MG195645	-
*Phyllachora sinobambusae*	China: Yunnan	MHYAU 085	*Sinobambusa tootsik* (Poaceae)	MG195655	MG195630	-
*Phyllachora sphaerocaryi* ^T^	China: Yunnan	MHYAU 178	*Sphaerocaryum malaccense* (Poaceae)	MK614114	MK614100	-
*Phyllachora sphaerocaryi*	China: Yunnan	MHYAU: 178	*Sphaerocaryum malaccense* (Poaceae)	-	MH018560	-
*Phyllachora thysanolaenae* ^T^	Thailand	MFLU 16-2071	*Thysanolaena maxima* (Poaceae)	-	-	MF372147
*Phyllachora vulgata voucher*	USA	BPI_640260	*Muhlenbergia mexicana* (Poaceae)	-	OP831215	-
*Phyllachora xinpingensis*	China: Yunnan	IFRD9465	*Chrysopogon aciculatus* (Poaceae)	OP359416	OP359398	-
*Phyllachora yuanjiangensis*	China: Yunnan	IFRD9466	*Arundinella setosa* (Poaceae)	OP359417	OP359399	OP359400
*Phyllachora yushaniae-falcatiauritae*	China: Yunnan	MHYAU 123	*Yushania falcatiaurita* (Poaceae)	MG195656	MG195631	-
*Phyllachora yushaniae-polytrichae*	China: Yunnan	MHYAU 122	*Yushania polytricha* (Poaceae)	MG195657	MG195632	MH992455
*Phyllachora yushaniae-polytrichae*	China: Yunnan	MHYAU 158	*Yushania polytricha* (Poaceae)	MG195658	MG195633	-
*Polystigma pusillum*	Costa Rica	MM-113	*Andira inermis* (Fabaceae)	KX430474	KX451907	KX451858
*Polystigma pusillum*	Costa Rica	MM-147	*Andira inermis* (Fabaceae)	KX430483	-	KX451862
*Polystigma pusillum*	Panama	MM-19	*Andira inermis* (Fabaceae)	KX430489	KX451899	KX451850
*Telimena bicincta*	Costa Rica	MM-108	*Picramnia antidesma* (Picramniaceae)	KX430473	KX451906	KX451857
*Telimena bicincta*	Costa Rica	MM-133	*Picramnia antidesma* (Picramniaceae)	KX430478	KX451910	KX451861

^a^ The newly generated sequences are indicated in bold. “^T^” marked ex-type or ex-epitype strains. ^b^ “-” means sequence unavailable.

## Data Availability

All sequence data are available in NCBI GenBank following the accession numbers in the manuscript. All species data are available in Index Fungorum.

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
