# Peer review of "Six Species of Phyllachora with Three New Taxa on Grass from Sichuan Province, China"

_jof, 2024, doi:10.3390/jof10080588_

Round 1

Reviewer 1 Report

This is a valuable taxonomic contribution to a poorly known group of fungi that have significant ecological and economic impacts. The new species described are well described and beautifully illustrated and also have good support both from morphological and DNA data, and the discussions on related species are valuable. There are, however, also shortcomings: the English has to be considerably improved and some technical/methodological issues have to be straightened out.

Review Phyllachora

51 What is ‘polyspores’? Reformulate.

101: pls provide a link to MAFFT

108: the models for Mr Bayes analyses (and how they were obtained) should be stated along with other basic settings.

133: ‘similar’ is not good enough; state the situation relating to conflicts. Please provide trees (e.g. as supplementary material) for each region and assess the situation for conflicts

148: would be better to order the species alphabetically

154. It is good that the authors have tried to supply statistical estimates for measurement, however they are inappropriate. This:  1.7 × 0.5 mm, n = 30’, has little meaning. The data is not testable for significance and the method for presentation measurements is not included in the Methods chapter. Estimates cannot be measured for compound entities, but only for individual ones; thus the width (if that is what it is?) of 1.7 should be replaced by some range around the arithm mean. M, based on the sd (standard deviation). For testing the significance among observations the number of observations has to be included, for every estimate. These comments hold true also for all ensuing measurements as long as they are suggested to be statistical estimates (ascomata, asci, spores etc). The measurements could for example be presented like this:   M-1 sd – M+1 sd (M=xx, sd= yy, n = number of obs) for each variable. Measurements indicated in the Abstract should as a consequence be revised.

169: please italics for Latin names

174: this would be a proper place to include a new table comparing all the species dealt with and other species relevant for comparisons, here e.g. P. thysanolaenae detailing important characteristics such as ascoma size, spore size etc

187: I have searched ref 25, but do not understand what ref 25 relates to? No Phyllochora is being mentioned. Why this ref?

192. Fig 2 is excellent, as are also the other photographic illustrations

407 That Phyllochoras are abundant in the area has not been shown in this paper. Pls remove.

412: What is an ‘obligate fungus’? and obligate fungi??? Can a fungus not be obligate??Incomprehensible. Explain.

Author Response

Response to Reviewer Comments

1. Summary

Thank you very much for taking the time to review this manuscript, and thank you very much for your suggestions on the manuscript. Please find the detailed responses below and the corresponding revisions. I have used red to indicate the modified part.

2. Questions for General Evaluation

Reviewer’s Evaluation

Response and Revisions

Does the introduction provide sufficient background and include all relevant references?

Yes

Are all the cited references relevant to the research?

Yes

Is the research design appropriate?

Yes

Are the methods adequately described?

Yes

Are the results clearly presented?

Yes

Are the conclusions supported by the results?

Yes

3. Point-by-point response to Comments and Suggestions for Authors

Comments 1: What is ‘polyspores’? Reformulate.

Response 1: Thank you for pointing this out. I apologize for not being clear about the meaning of “polyspores” in the text. I would like to express that in the asexual stage of Phyllachoraceae there is a large number of conidia in the ascospores. and has been updated in the manuscript in 54.

Comments 2:  pls provide a link to MAFFT

Response 2: Agree. I have done providing the link to MAFFT and added the login date in the manuscript 109.

Comments 3: the models for Mr Bayes analyses (and how they were obtained) should be stated along with other basic settings.

Response 3: Thank you for pointing this out. I agree with you and have modified it: ModelFinder was employed to select the best-fit partition model (Edge-linked) based on the BIC criterion. The best-fit model according to BIC was TIM2e+I+G4 for ITS, LSU, and SSU. Bayesian inference phylogenies were inferred using MrBayes under the partition model (2 parallel runs, 2,000,000 generations), within the initial 25% of sampled data discarded as burn-in, and the best nucleotide substitution model for each locus was identified using ModelFinder of Phylosuite. The best-fit model according to AIC was SYM+FQ+G4 for ITS, GTR+F+G4 for LSU, and TN+F+G4 for SSU, respectively. And has been updated in the manuscript in 117-122.

Comments 4: ‘similar’ is not good enough; state the situation relating to conflicts. Please provide trees (e.g. as supplementary material) for each region and assess the situation for conflicts

Response 4: I apologize for the misunderstanding due to my lack of clarity, and I have corrected it to read ‘Since the ML and BI phylogenetic trees exhibited similar topologies, only the ML tree (Figure 1) is presented’. And has been updated in the manuscript in 146-148.

Comments 5: would be better to order the species alphabetically

Response 5: Thank you for pointing this out. In the manuscript I have order the species alphabetically.

Comments 6: It is good that the authors have tried to supply statistical estimates for measurement, however they are inappropriate. This: ‘1.7 × 0.5 mm, n = 30’, has little meaning. The data is not testable for significance and the method for presentation measurements is not included in the Methods chapter. Estimates cannot be measured for compound entities, but only for individual ones; thus the width (if that is what it is?) of 1.7 should be replaced by some range around the arithm mean. M, based on the sd (standard deviation). For testing the significance among observations the number of observations has to be included, for every estimate. These comments hold true also for all ensuing measurements as long as they are suggested to be statistical estimates (ascomata, asci, spores etc). The measurements could for example be presented like this:   M-1 sd – M+1 sd (M=xx, sd= yy, n = number of obs) for each variable. Measurements indicated in the Abstract should as a consequence be revised.

Response 6: Thank you for pointing this out. I understand what you are saying and am seriously considering your suggestion. However, the size differences between mature ascomata, ascospores, asci, etc. of Phyllachora species are not large enough to make calculations using the squared difference formula meaningful, and we have followed the descriptions of studies on Phyllachora species in other articles. In the manuscript, both compound units and individual ascomatas of Phyllachora species were measured. The data measured for each species in the manuscript show the maximum value, minimum value, average value, and the number of measurements taken, with a maximum of 50 measurements, and it is also mentioned in the Materials and Methods section that the measurements were made using the Tarosoft® Image Framework (version 0.9.7) software developed by Tarosoft(R) in Nonthaburi, Thailand.

Comments 7: 169: please italics for Latin names

Response 7: Thank you for pointing this out. I will carefully check the problems and correct. And has been updated in the manuscript in 231.

Comments 8: 174: this would be a proper place to include a new table comparing all the species dealt with and other species relevant for comparisons, here e.g. P. thysanolaenae detailing important characteristics such as ascoma size, spore size etc

Response 8: Thank you very much for your suggestion. It is good to find a proper place to include a new table comparing all the species dealt with and other species related for comparisions, which can make the article clearer. But after my careful consideration, I think it is inappropriate to add a new table.There are mainly the following reasons: 1. The content in the form may be repeated with the content of the Notes; 2. The number of new species in the manuscript is not large, and it will feel very abrupt to add to the form; 3. It would be better to explain the differences between the new species and other species in the manuscript separately in the Notes section. 

Comments 9: 187: I have searched ref 25, but do not understand what ref 25 relates to? No Phyllochora is being mentioned. Why this ref?

Response 9: Thank you for pointing this out. The significance of reference 25 is to provide how to identify new species. Although Phyllochora is not mentioned in the reference, it has good reference value for identifying a new species, so I added it to the manuscript

Comments 10: 192. Fig 2 is excellent, as are also the other photographic illustrations

Response 10: Thank you very much, we are very happy to get your affirmation.

Comments 11: 407 That Phyllochoras are abundant in the area has not been shown in this paper. Pls remove.

Response 11: Thank you for pointing this out. I will carefully check the problems and correct. And has been updated in the manuscript in 231.

Comments 12: What is an ‘obligate fungus’? and obligate fungi??? Can a fungus not be obligate??Incomprehensible. Explain.

Response 12: : Based on a large number of previous studies on Phyllachora species, it can be seen that Phyllachora species are a class of obligate fungi, which means that Phyllachora species are host-plant specific. Therefore, previous identification studies of Phyllachora species were mostly based on the host plant species, which was inaccurate, and several examples are used in the manuscript to prove this. See lines 438-443 of the manuscript for details.

Reviewer 2 Report

See the observations reported in the “Detail comments” section.

If I correctly read the paper jof-3102156 “Six Species of Phyllachora with Three New Taxa on Grass from Sichuan Province, China”, the authors report data on the genus Phyllachora (Fungi, Ascomycota, Pezizomycotina, Sordariomycetes, Phyllachorales, Phyllachoraceae) and its species causing tar spots on plants. Six species collected from grasses (Poaceae) in the Sichuan Province of China were described: the well-known P. chloridis, P. graminis, and P. miscanthi, and three novel species P. chongzhouensis, P. huiliensis, and P. neidongensis. The six species were characterized on morphological features and differentiated by host relationship and the phylogenetic analyses combining the ribosomal internal transcribed spacer (ITS), the large nuclear ribosomal RNA subunit (LSU), and the small subunit ribosomal ribonucleic acid (SSU) loci. Multigene analyses were based on 90 GenBank accessions belonging to 50 species included in six genera (Ascovaginospora, Camarotella, Coccodiella, Neophyllachora, Phyllachora, and Polystigma) within the Phyllachoraceae family. Two strains of Telimena bicincta were chosen as the outgroups. The accession used came from different botanical species and countries. Morphological features considered asci, asci walls, ascospores, and paraphyses.

Notwithstanding the scientific sound of this work, the presentation in the form of a manuscript lacks scientific strictness. It is difficult to read, confusing, unclear, and requires adjustments. Follow the “Journal of Fungi” guidelines and template (https://www.mdpi.com/journal/jof/instructions) to improve the manuscript.

 Following the practice adopted by the International Code of Nomenclature for Algae, Fungi and Plants scientific names of all taxonomic ranks should be italicized. In manuscripts dealing with taxonomy, for every organism, the full genus name and authority of the genus or species should be included at first mention. For manuscripts dealing with subjects other than taxonomy, this is desirable but not essential. The genus name should be abbreviated to the initial letter if no ambiguity arises, although the full genus name should always be used.

Insert the Author’s name into all the cited organisms.

Pay attention to punctuation and typing errors. E.g.:

            Line 25: insert a space between “10” and “µm”.

            Line 27: separate “:” and “3”.

            Line 50: insert a space between “sheath” and “[3,”.

            Line 99: use “Table 1” instead of “Table 1.”.

            Line 104: insert a space between “1.2.3.” and “[21”.

            Lines 203, 245, 280, 320, 361, 425, 427: What does “tarspot” mean?

In the text, use the acronym after the definition.

Delete unutilized abbreviations (e.g., MLZ in line 78).

The acronym “nrLSU” was used in the “abstract” section and “LSU” in the text. Please uniform.

Avoid repetition of concepts.

 The title is not completely pertinent to the topics of the manuscript. Improve.

Is the description of P. chloridis, P. graminis, and P. miscanthi necessary?

 A possible suggestion: "Phyllachora chongzhouensis, P. huiliensis, and P. neidongensis novel species on Grasses (Poaceae) in Sichuan Province (China)

or Phyllachora Species on Grasses (Poaceae) in Sichuan Province (China): the well know P. chloridis, P. graminis, and P. miscanthi, and three novel species P. chongzhouensis, P. huiliensis, and P. neidongensis”

The abstract is a very important part of the paper, after reading the manuscript I suggest rewriting it.

Use pertinent keywords. E.g., Tar spot fungi, mycodiversity.

Line 27: use “three” instead of “3”.

Line 27: use “ Multigene phylogeny” instead of “phylogeny”.

Line 27: What does “Poacea” mean? Use “Poaceae

 The “Introduction” section is not clear. Improve.

Line 38: What does “ML” mean?

Line 38: What does “BI” mean?

Line 41: What does “Fri.” mean?

Introduce some examples of botanical species host of tar spot fungi.

Improve symptoms and signs description.

Lines 55-56: period not clear. Rewrite.

Lines 56-62: periods not clear. Rewrite.

The “Materials and Methods” section is not clear. Improve.

This section should describe the step-by-step applied protocols to allow others to replicate the published results.

If it is possible insert the survey period and plant host sampled in Section 2.1.

Line 64: use “Specimens Collection and Herbarium Deposit” instead of “Specimen collection”.

Line 70: use “Studies” instead of “Observations”.

Line 84: use “DNA Extraction, Amplification, and Sequencing” instead of “DNA Extraction, PCR, and Sequencing”.

Lines 86-87: use “with the manufacturer’s instructions” instead of “to the manual”.

Line 95: use “China, using forward and reverse primers.” instead of “China.”.

Line 96: use “Sequence Alignment and Phylogenetic Analyses” instead of “Phylogenetic Analyses”.

Line 97: delete “using forward and reverse primers”.

Line 108: delete “, respectively”.

Tables and Figures must be self-explanatory, clear, and easy to understand without extra explanation. Improve.

Table 1:

What does “C. calatheae” mean? Camarotella calatheae or Coccodiella calatheae.

To avoid ambiguity among the abbreviated genus names (e.g., Camarotella and Coccodiella, Phyllachora and Polystigma) use the full genus name.

Follow the “Journal of Fungi” guidelines and template, I suggest the footers.

Lines 113-115: periods not clear. Rewrite.

A possible suggestion:

Table 1. Samples used for multigene phylogenetic analysis. a,T

Use “Strain” instead of “Strain number”.

Use “Host (Family)” instead of “Host”.

 Insert “b” near the “-” under the ITS column.

footers

a Sequences generated in this study are indicated in bold.

b Sequences unavailable.

T marked ex-type or ex-epitype strains.

The whole “Results” section is not clear. Enhance.

Avoid information on “material and methods”.

 Why were the Camarotella and Coccodiella genera considered as Camarotella?

 Figure 1:

Improve magnification

Insert Coccodiella section.

Lines 124-129: periods not clear. Rewrite

Line 128: What does “study species” mean?

Lines 132-146: periods not clear. Rewrite.

Line 132: What does “new collections” mean?

Line 142: What does “our collection” mean?

Improve Section 3.2. Taxonomy

Is the description of P. chloridis, P. graminis, and P. miscanthi necessary?

Line 169: write “Phragmites australis” in italics.

Line 175: use “P.” instead of “Phyllachora”.

 Lines 194, 218, 238, 314-315, 336-337, 353-354: What does “Vertical section” mean? I suggest “Vertical cross-section”, “transverse section” or “cross-section”.

 Line 228: What does “our collections” mean? May be: SICAU 24-0054 and SICAU 240053.

 Line 231: insert a citation after “species”.

 Improve the “Discussion” section.

 Arrange “Reference” following the “Journal of Fungi” guidelines and template (https://www.mdpi.com/journal/jof/instructions).

 Kind regards

Author Response

Response to Reviewer Comments

1. Summary

Thank you very much for taking the time to review this manuscript, and thank you very much for your suggestions on the manuscript. Please find the detailed responses below and the corresponding revisions. I have used red to indicate the modified part.

2. Questions for General Evaluation

Reviewer’s Evaluation

Response and Revisions

Does the introduction provide sufficient background and include all relevant references?

Yes

Are all the cited references relevant to the research?

Yes

Is the research design appropriate?

No

The title is inappropriate.

Are the methods adequately described?

Yes

Are the results clearly presented?

Yes

Are the conclusions supported by the results?

Yes

3. Point-by-point response to Comments and Suggestions for Authors

Comments 1: Insert the Author’s name into all the cited organisms.

Pay attention to punctuation and typing errors. E.g.:

(1) Line 25: insert a space between “10” and “µm”.

(2) Line 27: separate “:” and “3”.

(3) Line 50: insert a space between “sheath” and “[3,”.

(4) Line 99: use “Table 1” instead of “Table 1.”.

(5) Line 104: insert a space between “1.2.3.” and “[21”.

Response 1: Thank you for pointing this out. I will carefully check the problems and correct them. Please see attached for details.

Comments 2: Lines 203, 245, 280, 320, 361, 425, 427: What does “tarspot” mean?

Response 2: I apologize for the misunderstanding due to my lack of clarity. I've changed “tarspot” to “tar spot” to indicate a manifestation of Phyllachora when infects plants.

Comments 3: (1) In the text, use the acronym after the definition. Delete unutilized abbreviations (e.g., MLZ in line 78).

(2)The acronym “nrLSU” was used in the “abstract” section and “LSU” in the text. Please uniform. Avoid repetition of concepts.

Response 3: Thank you for pointing this out. I will carefully check the problems and correct them.

Comments 4: (1)The title is not completely pertinent to the topics of the manuscript. Improve.

(2)Is the description of P. chloridis, P. graminis, and P. miscanthi necessary?

Response 4: (1) Thank you for pointing this out. I'm all in favor of it. At your suggestion I revised the title of the manuscript to: Phyllachora Species on Grasses (Poaceae) in Sichuan Province (China):Three new species and three redescribed taxa 

(2)The question of whether it is necessary to describe the three known species of P. chloridis, P. graminis, and P. miscanthi. I think it's important. In reviewing the original references for the descriptions of P. chloridis, P. graminis, and P. miscanthi, I found that the references for their descriptions are relatively old, and some of the descriptions are in Chinese or abbreviated. In addition, the illustrations were simple, and many structures were not shown. In the molecular identification, the ITS sequence is short, and the ITS,LSU,SSU sequences are incomplete. Therefore, the redescription of P. chloridis, P. graminis, and P. miscanthi is important and can provide important references for other authors to follow.

Comments 5: The abstract is a very important part of the paper, after reading the manuscript I suggest rewriting it.

Response 5: Agree. I've changed the abstract: Phyllachora (Phyllachoraceae, Phyllachorales) species are parasitic fungi with a wide global distribution, causing tar spots on plants. In this study, we described three newly discovered species: Phyllachora chongzhouensis, Phyllachora neidongensis, and Phyllachora huiliensis, as well as three known species of this genus from Poaceae in China. These species were characterized using morphological traits and multi-locus phylogeny based on the internal transcribed spacer region (ITS) with intervening 5.8S rRNA gene, the large subunit of the rRNA gene (LSU), and the 18S ribosomal RNA gene (SSU). The study also provides further evidence for the identification of these taxa.

Comments 6: Use pertinent keywords. E.g., Tar spot fungi, mycodiversity.

Line 27: use “three” instead of “3”.

Line 27: use “ Multigene phylogeny” instead of “phylogeny”.

Line 27: What does “Poacea” mean? Use “Poaceae”

Response 6: Thank you for pointing this out. I will carefully check the problems and correct them. Please see attached for details.

Comments 7: The “Introduction” section is not clear. Improve.

Response 7: Thank you for pointing this out. I have further refined. Please see attached for details.

Comments 8: Line 38: What does “ML” mean?

Line 38: What does “BI” mean?

Line 41: What does “Fri.” mean?

Response 8: I apologize for the misunderstanding due to my lack of clarity. “ML” means ”Maximum likelihood”, “BI” measn “Bayesian inference”, “Fri.” amend “May”. I've revised the 38, and 49 lines in the manuscript.

Comments 9: Improve symptoms and signs description.

Lines 55-56: period not clear. Rewrite.

Lines 56-62: periods not clear. Rewrite.

Response 9: Thank you very much for your suggestions, I have been carefully revised in the manuscript lines 60 - 68.

Comments 10: The “Materials and Methods” section is not clear. Improve.

This section should describe the step-by-step applied protocols to allow others to replicate the published results.

If it is possible insert the survey period and plant host sampled in Section 2.1.

Line 64: use “Specimens Collection and Herbarium Deposit” instead of “Specimen collection”.

Line 70: use “Studies” instead of “Observations”.

Line 84: use “DNA Extraction, Amplification, and Sequencing” instead of “DNA Extraction, PCR, and Sequencing”.

Lines 86-87: use “with the manufacturer’s instructions” instead of “to the manual”.

Line 95: use “China, using forward and reverse primers.” instead of “China.”.

Line 96: use “Sequence Alignment and Phylogenetic Analyses” instead of “Phylogenetic Analyses”.

Line 97: delete “using forward and reverse primers”.

Line 108: delete “, respectively”.

Response 10: Thank you for pointing this out. I agree with you and have modified it. Please see attached for details.

Comments 11: Table 1:

What does “C. calatheae” mean? Camarotella calatheae or Coccodiella calatheae.

To avoid ambiguity among the abbreviated genus names (e.g., Camarotella and Coccodiella, Phyllachora and Polystigma) use the full genus name.

Follow the “Journal of Fungi” guidelines and template, I suggest the footers.

Lines 113-115: periods not clear. Rewrite.

A possible suggestion:

Table 1. Samples used for multigene phylogenetic analysis. a,T

Use “Strain” instead of “Strain number”.

Use “Host (Family)” instead of “Host”.

Insert “b” near the “-” under the ITS column.

footers

a Sequences generated in this study are indicated in bold.

b Sequences unavailable.

T marked ex-type or ex-epitype strains

Response 11: Thank you for pointing this out. I have made the changes you suggested. See the attachment for details.

Comments 12: The whole “Results” section is not clear. Enhance.

Avoid information on “material and methods”.

Response 12: Thank you very much for your suggestions, I have been carefully revised in the manuscript lines 139 -160, please see the attachment for details.

Comments 13: Why were the Camarotella and Coccodiella genera considered as Camarotella?

Response 13: Thank you for pointing this out. Camarotella and Coccodiella are two separate genera, and I apologize for my carelessness in grouping Camarotella and Coccodiella into one genus, which I have corrected in the manuscript.

Comments 14: Figure 1:

Improve magnification

Insert Coccodiella section.

Lines 124-129: periods not clear. Rewrite

Line 128: What does “study species” mean?

Response 14: Thank you for pointing this out. I have insert Coccodiella section in the manuscript and rewrite the contents of lines 130-135.

“study species” means: SICAU 24-0053, SICAU 24-0054, SICAU 24-0044, SICAU 24-0045, SICAU 24-0051, SICAU 24-0052, SICAU 24-0048, SICAU 24-0049, SICAU 24-0050, SICAU 24-0046, SICAU 24-0047.

Comments 15: Lines 132-146: periods not clear. Rewrite.

Line 132: What does “new collections” mean?

Line 142: What does “our collection” mean?

Response 15: Very much your advice and pointing this out. I will carefully revise this part, please see attached.

“new collections” means new species in the manuscript: SICAU 24-0044, SICAU 24-0045,  SICAU 24-0048, SICAU 24-0049, SICAU 24-0046, SICAU 24-0047

“our collections” means known species in the manuscript:SICAU 24-0053, SICAU 24-0054, SICAU 24-0051, SICAU 24-0052, SICAU 24-0050

Comments 16: Is the description of P. chloridis, P. graminis, and P. miscanthi necessary?

Response 16: Thank you for pointing this out. I think it's important. In reviewing the original references for the descriptions of P. chloridis, P. graminis, and P. miscanthi, I found that the references for their descriptions are relatively old, and some of the descriptions are in Chinese or abbreviated. In addition, the illustrations were simple, and many structures were not shown. In the molecular identification, the ITS sequence is short, and the ITS,LSU,SSU sequences are incomplete. Therefore, the redescription of P. chloridis, P. graminis, and P. miscanthi is important and can provide important references for other authors to follow.

Comments 17: Line 169: write “Phragmites australis” in italics.

Line 175: use “P.” instead of “Phyllachora”.

Lines 194, 218, 238, 314-315, 336-337, 353-354: What does “Vertical section” mean? I suggest “Vertical cross-section”, “transverse section” or “cross-section”.

Response 17: Thank you for pointing this out. I agree with you and have modified it.

Comments 18: Line 228: What does “our collections” mean? May be: SICAU 24-0054 and SICAU 240053.

Response 18: Thank you for pointing this out. “our collections” means SICAU 24-0054 and SICAU 240053.

Comments 19:  Line 231: insert a citation after “species”.

Improve the “Discussion” section.

Arrange “Reference” following the “Journal of Fungi” guidelines and template (https://www.mdpi.com/journal/jof/instructions).

Response 19: Thank you very much for your suggestions, I have been carefully revised in the manuscript, please see the attachment for details.

Round 2

Reviewer 2 Report

See the observations reported in the “Detail comments” section.

Abstract and line 63: Add information of response 16 “. In reviewing the original references of P. chloridis, P. graminis, and P. miscanthi are relatively old, and in Chinese or abbreviated. In addition, the illustrations were simple, and many structures were not shown. In molecular identification, the ITS sequence is short, while the ITS, LSU, SSU are incomplete. Therefore, the redescription of P. chloridis, P. graminis, and P. miscanthi provides new important references.””

Lines 54-57: the sentence “The asexual manifestations of Phyllachoraceae contain a large number of conidia in the ascospores, are hyaline, more elongated, and are recognized as coelomycetes, demonstrating either spermatic or disseminative properties [10-12].” is not clear. Improve.

Line 55: What does “contain a large number of conidia in the ascospores” mean?

Are the conidia inside the ascospores?

Line 58: What does “plants” mean? All plants? Or Host plants

Line 113: put “Telimena bicincta” in italics.

Table 1:

Lines 126-127: use “Table 1. Samples used for multigene phylogenetic analysis. a

Use “Numbers b” instead of “Number”

Insert the border at the end of the table

Use: “a The newly generated sequences are indicated in bold.“T” marked ex-type or ex-epitype strains.” Instead of “The newly generated sequences are indicated in bold.“T” marked ex-type or ex-epitype strains.”

Use: “b “-” means sequence unavailable.” instead of ““-” means sequence unavailable”

Line 146: use “SICAU 24-0044, SICAU 24-0045,  SICAU 24-0048, SICAU 24-0049, SICAU 24-0046, SICAU 24-0047” instead of “new collections”.

Line 156: delete “our collections (” and “)”

Line 194: explain “our collections”

Lines 285, 290, 356 and 362 and: delete “our collections (” and “)”

Lines 331-332: replace “our collections” with the specie acronym considered.

Lines 413-414: use “Therefore, P. neidongensis strain SICAU 24-0046 was proposed as a new species.” Instead of “Therefore …species.”.

Line 442 delete “our”

Arrange the “Reference” section following the “Journal of Fungi” guidelines and template (https://www.mdpi.com/journal/jof/instructions).

1.       Author 1, A.B.; Author 2, C.D. Title of the article. Abbreviated Journal Name Year, Volume, page range.

2.       Author 1, A.; Author 2, B. Title of the chapter. In Book Title, 2nd ed.; Editor 1, A., Editor 2, B., Eds.; Publisher: Publisher Location, Country, 2007; Volume 3, pp. 154–196.

3.       Author 1, A.; Author 2, B. Book Title, 3rd ed.; Publisher: Publisher Location, Country, 2008; pp. 154–196.

4.       Author 1, A.B.; Author 2, C. Title of Unpublished Work. Abbreviated Journal Name year, phrase indicating stage of publication (submitted; accepted; in press).

5.       Author 1, A.B. (University, City, State, Country); Author 2, C. (Institute, City, State, Country). Personal communication, 2012.

6.       Author 1, A.B.; Author 2, C.D.; Author 3, E.F. Title of Presentation. In Proceedings of the Name of the Conference, Location of Conference, Country, Date of Conference (Day Month Year).

7.       Author 1, A.B. Title of Thesis. Level of Thesis, Degree-Granting University, Location of University, Date of Completion.

8.       Title of Site. Available online: URL (accessed on Day Month Year).

Author Response

Response to Reviewer Comments

1. Summary

Thank you very much for taking the time to review this manuscript, and thank you very much for your suggestions on the manuscript. Please find the detailed responses below and the corresponding revisions. I have used red to indicate the modified part. In addition, we have modified the "References" section in accordance with the "Journal of Mycology" guidelines and template (https://www.mdpi.com/journal/jof/instructions).

2. Questions for General Evaluation

Reviewer’s Evaluation

Response and Revisions

Does the introduction provide sufficient background and include all relevant references?

Yes

Are all the cited references relevant to the research?

Yes

Is the research design appropriate?

Yes

Are the methods adequately described?

Yes

Are the results clearly presented?

Yes

Are the conclusions supported by the results?

Yes

3. Point-by-point response to Comments and Suggestions for Authors

Comments 1: Abstract and line 63: Add information of response 16 “. In reviewing the original references of P. chloridis, P. graminis, and P. miscanthi are relatively old, and in Chinese or abbreviated. In addition, the illustrations were simple, and many structures were not shown. In molecular identification, the ITS sequence is short, while the ITS, LSU, SSU are incomplete. Therefore, the redescription of P. chloridis, P. graminis, and P. miscanthi provides new important references.”

Response 1: Thank you for pointing this out and I have corrected it in the abstract and line 63 of the manuscript.

Comments 2: Lines 54-57: the sentence “The asexual manifestations of Phyllachoraceae contain a large number of conidia in the ascospores, are hyaline, more elongated, and are recognized as coelomycetes, demonstrating either spermatic or disseminative properties [10-12].” is not clear. Improve.

Line 55: What does “contain a large number of conidia in the ascospores” mean?

Are the conidia inside the ascospores?

Response 2: I apologize for the misunderstanding due to my lack of clarity, and I have corrected it to read “The asexual manifestations of Phyllachoraceae has been reported as a coelomycetous morph, demonstrating either spermatic or disseminative properties”. And has been updated in the manuscript in 54-57.

Comments 3: Line 58: What does “plants” mean? All plants? Or Host plants

Response 3: Thank you for pointing this out. “plants” means Host plants, I have corrected it in the manuscript.

Comments 4: Line 113: put “Telimena bicincta” in italics.

Response 4: Thank you for pointing this out and I have corrected it in the manuscript in 133.

Comments 5: Table 1:

Lines 126-127: use “Table 1. Samples used for multigene phylogenetic analysis. a

Use “Numbers b” instead of “Number”

Insert the border at the end of the table

Use: “a The newly generated sequences are indicated in bold.“T” marked ex-type or ex-epitype strains.” Instead of “The newly generated sequences are indicated in bold.“T” marked ex-type or ex-epitype strains.”

Use: “b “-” means sequence unavailable.” instead of ““-” means sequence unavailable”

Response 5: Thank you for pointing this out. I agree with you and have modified it.

Comments 6: Line 146: use “SICAU 24-0044, SICAU 24-0045,  SICAU 24-0048, SICAU 24-0049, SICAU 24-0046, SICAU 24-0047” instead of “new collections”.

Response 6: Thank you for pointing this out. I agree with you and have modified it.

Comments 7: Line 156: delete “our collections (” and “)”

Response 7: Thank you for pointing this out. I agree with you and have modified it.

Comments 8: Line 194: explain “our collections”

Response 8: Thank you for pointing this out. “Our collection” means SICAU 24-0053 and SICAU 24-0054 , which in the manuscript I have changed to SICAU 24-0053 and SICAU 24-0054.

Comments 9: Lines 285, 290, 356 and 362 and: delete “our collections (” and “)”

Response 9: Thank you for pointing this out. I agree with you and have modified it.

Comments 10: Lines 331-332: replace “our collections” with the specie acronym considered.

Response 10: Thank you for pointing this out. I agree with you and have modified it.

Comments 11: 4Lines 413-414: use “Therefore, P. neidongensis strain SICAU 24-0046 was proposed as a new species.” Instead of “Therefore …species.”.

Response 11: Thank you for pointing this out and I have corrected it in the manuscript in 413.

Comments 12: Line 442 delete “our”

Response 12: : Thank you for pointing this out and I have corrected it in the manuscript in 442.
